# Evaluation of Water Vapor Radiative Effects Using GPS Data Series over Southwestern Europe

**Javier Vaquero-Martínez** [1,2,*] [ID]**, Manuel Antón** [1,2]**, Arturo Sanchez-Lorenzo** [1,2]
**and Victoria E. Cachorro** [3] [ID]

1   Departamento de Física, Universidad de Extremadura, CP 06006 Badajoz, Spain; mananton@unex.es (M.A.);
    arturosl@unex.es (A.S.-L.)
2   Instituto Universitario de Investigación del Agua, Cambio Climático y Sostenibilidad (IACYS),
    Universidad de Extremadura, CP 06006 Badajoz, Spain
3   Grupo de Óptica Atmosférica, Universidad de Valladolid, Paseo Belén 7, CP 47011 Valladolid, Spain;
    chiqui@goa.uva.es
*   Correspondence: javier_vm@unex.es

**Abstract:** Water vapor radiative effects (WVRE) at surface in the long-wave (LW) and short-wave (SW) spectral ranges under cloud and aerosol free conditions are analyzed for seven stations in Spain over the 2007–2015 period. WVRE is calculated as the difference between the net flux obtained by two radiative transfer simulations; one with water vapor from Global Positioning System (GPS) measurements and the other one without any water vapor (dry atmosphere). The WVRE in the LW ranges from 107.9 Wm$^2$ to 296.7 Wm$^{-2}$, while in the SW it goes from $-64.9$ Wm$^{-2}$ to $-6.0$ Wm$^{-2}$. The results show a clear seasonal cycle, which allows the classification of stations in three sub-regions. In general, for total (SW + LW) and LW WVRE, winter (DJF) and spring (MAM) values are lower than summer (JJA) and autumn (SON). However, in the case of SW WVRE, the weaker values are in winter and autumn, and the stronger ones in summer and spring. Positive trends for LW (and total) WVRE may partially explain the well-known increase of surface air temperatures in the study region. Additionally, negative trends for SW WVRE are especially remarkable, since they represent about a quarter of the contribution of aerosols to the strong brightening effect (increase of the SW radiation flux at surface associated with a reduction of the cloud cover and aerosol load) observed since the 2000s in the Iberian Peninsula, but with opposite sign, so it is suggested that water vapor could be partially masking the full magnitude of this brightening.

**Keywords:** water vapor; radiative effect; long-wave; Spain; Southwestern Europe; Europe;

## 1. Introduction

Water vapor is acknowledged as a crucial element in the climate system. Its latent heat has an important role in energy transport and it is obviously a fundamental part of the hydrological cycle [1]. Water vapor is also known to be the main absorber of the infrared radiation emitted by Earth's surface and atmosphere, which allows heating of the low atmosphere. Its hydroxyl (H–O) bond is the cause of the absorption in the infrared region.

The infrared absorption of water vapor involves a positive feedback [2,3]. If the atmosphere's temperature rises, the air can hold more water vapor from evaporation, since the saturation vapor pressure increases as temperature rises. This further increases the temperature of the climate system because of water vapor heating. Therefore, the effect of water vapor is considered a feedback rather than a forcing, since, on a global scale, the water vapor concentration is mainly dependent on

the temperature, and the typical residence time of water vapor is about ten days [1]. Hence, the anthropogenic emissions of water vapor have no significant effect on the global climate, except those in the stratosphere, where due to the conditions of stability, pressure and temperature, water vapor emissions manage to stay in the long term, and therefore can be considered a forcing [4–7].

Water vapor is not evenly distributed in the atmospheric column. The lower layers of air generally hold most of the water vapor, sharply decreasing with height. Therefore, water vapor is generally quantified using the column integrated amount of water vapor or integrated water vapor (IWV). This is equivalent to condensing all the water vapor in the atmospheric column and measuring the height that it would reach in a vessel of unit cross section. The units are those of columnar mass density ($gcm^{-2}$ or $kgm^{-2}$). Since 1 g of liquid water has a volume of 1 $cm^3$, the columnar mass density can be written as units of length (1 cm).

However, there is a great uncertainty about the quantification of the radiative effects of water vapor. Although some efforts have been made to study it in the short-wave (SW) region (i.e., [8–16]), the references in the literature related to the LW effects of water vapor are scarce (i.e., [17–21]). These works mainly focus in the feedback and sensitivity of climate system with respect to water vapor and the downwelling long-wave (LW) radiation, but not considering the radiative balance. This work aims to shed some light on the role of water vapor in the radiative balance, not only in SW as previous studies have done, but also in LW. The implications of this work can help to understand the trends in solar radiation at surface, as well as the increase of temperatures in the Iberian Peninsula in the study period. Similar approaches have been conducted with other atmospheric compounds, like aerosols (i.e., [22]), aerosols and clouds [23,24], ozone [25,26], even stratospheric water vapor [7].

In this work, the water vapor radiative effect (WVRE) is defined as the net change in radiation at surface taking as reference a dry atmosphere (adapted from [9]). Daily values (calculated as the integration of hourly values) of WVRE in both SW and LW regions are presented for GPS stations in Spain, for the period 2007–2015. The total WVRE is also analyzed, obtained as the sum of the SW and LW effects. The paper has the following structure: Section 2 describes the data-sets used in this work and in Section 3 the methodology is explained. The results are exhibited in Section 4, while the discussion of results is carried out in Section 5. Finally, conclusions are drawn in Section 6.

## 2. Data

### 2.1. IWV Data from GPS stations

Global Positioning System (GPS) ground-based stations can be used to measure IWV as thoroughly detailed in Bevis et al. [27]. In short, in GPS positioning, the troposphere causes a delay in the signal between the GPS satellite and the GPS receiver that must be estimated for precise positioning, since such delay is of the order of a few meters. The direct measurement of this delay is known as Slant Tropospheric Delay (STD). STD is the result of two contributions, one related to the non-dipolar contribution that all gases in the atmosphere cause (known as Slant Hydrostatic Delay, SHD), and another contribution due to the dipolar effects in water vapor molecules, which is known as Slant Wet Delay (SWD). The sum of both contributions gives the STD, as Equation (1) shows.

$$STD = SWD + SHD \qquad (1)$$

However, the slant delays change with the geometry. They depend on the angle between the satellite–receiver line and the zenith. Therefore, a mapping function [28–30] is needed to convert slant delays to zenith delays, as shown in Equation (2).

$$STD = m_w(E)ZWD + m_h(E)ZHD = m(E)(ZWD + ZHD) = m(E)ZTD \qquad (2)$$

where $E$ is the elevation angle, $m$ are the mapping functions, ZTD is the Zenith Tropospheric Delay, ZWD the Zenith Wet Delay and ZHD the Zenith Hydrostatic Delay.

ZTD is provided by GPS processing methods. ZHD can be obtained using a simple model [31] based on the surface pressure. The difference between ZTD and ZHD gives ZWD, which can be converted to IWV with a conversion factor $ZWD = \kappa \cdot IWV$. The constant $\kappa$ can be determined from the mean air temperature of the atmospheric column weighted by the water vapor content. This mean air temperature is estimated from an empirical linear relationship with temperature at the station level.

The period covered in this work ranges from 2007 to 2015, since this is the time span of available GPS data. The tropospheric products are provided by the Spanish Geographic Institute "Instituto Geográfico Nacional", which is a local analysis center for the European Reference Frame (EUREF). The stations selected are those that meet the quality standards for EUREF network, have a long uninterrupted time-series, have nearby meteorological automatic stations and have a representative number of cloud-free days in all (or most) months of the year. Surface pressure and temperature were provided by the Spanish Meteorological State Agency (AEMet). Temperature is provided hourly, while pressure is measured four times a day. Temperature was linearly interpolated to the time of measurements, and pressure was interpolated taking into account the barometric tide. This is done for the seven stations shown in Figure 1. A summary of the stations positions is presented in Table 1. The temporal resolution of the IWV data-set is one hour.

**Table 1.** Position of the GPS stations. Latitude and longitude are given in degrees north and east, while altitude is given in meters. Zones are: North atlantic (NA), Mediterranean Sea (MS) and interior (I).

| Station | Acronym | Latitude | Longitude | Altitude | Zone |
|---------|---------|----------|-----------|----------|------|
| A Coruña | acor | 43.36 | −8.40 | 12 | NA |
| Córdoba | coba | 37.92 | −4.72 | 162 | MS |
| Villafranca | vill | 40.44 | −3.95 | 596 | I |
| Alicante | alac | 38.34 | −0.48 | 10 | MS |
| Almería | alme | 36.85 | −2.46 | 77 | MS |
| Valencia | vale | 39.48 | −0.34 | 28 | MS |
| Cáceres | cace | 39.48 | −6.34 | 384 | I |

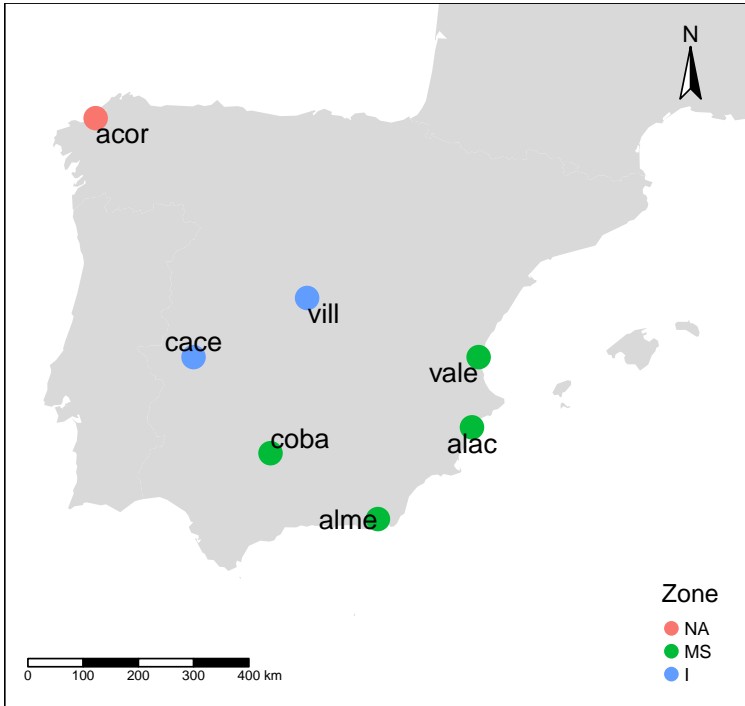

**Figure 1.** Map of the Global Positioning System (GPS) stations included in this work. Zones are: North atlantic (NA), Mediterranean Sea (MS) and interior (I).

## 2.2. Auxiliary Data

Some additional data sets were used in order to characterize the state of the atmosphere and surface. Profiles of temperature, ozone, and water vapor were obtained from ERA-Interim reanalysis. The latter were re-scaled to the IWV value from GPS stations. The temperature of the surface was also obtained from ERA-Interim reanalysis for LW calculations.

The ERA-Interim [32] is the latest global reanalysis model produced by the European Centre for Medium-Range Weather Forecasts (ECMWF) to replace ERA-40. The data is available at 4 times a day (00, 06, 12, 18 h). For every IWV measurement the nearest pixel at the closest time is selected to represent the state of the atmosphere and surface. The ERA-Interim grid has a resolution of $0.75° \times 0.75°$ with 37 vertical levels.

Additionally, daily sunshine duration records and cloud cover (CC) data provided by AEMet are also used for the selection of days with cloud free and low aerosol load conditions.

## 3. Methodology

Both LW and SW irradiances have been simulated using the Santa Barbara's DIScrete Ordinate Radiative Transfer (DISORT) Radiative Transfer model (SBDART, [33]), only for those days considered as cloud and aerosol free (Rayleigh atmosphere). For that, firstly, days with CC less than or equal to 1 okta are selected. Subsequently, from these cloud-free days, to select cases with low aerosol load, daily sunshine duration is divided by its theoretical value under the assumption of cloud-free sky, and a threshold value of 0.75 is used to filter out heavy aerosol load situations. WMO [34] recommends the value of 0.70, being 0.75 a more restrictive threshold to ensure aerosol free days.

The SW wavelength range considered was between 0.2 μm and 4.0 μm (0.5% steps, ranging from 0.001 to 0.02 μm). More detailed description on the variable inputs to the SW simulations can be found in Vaquero-Martínez et al. [13]. The atmosphere model were "mid-latitude summer" from March to August (both included) and "mid-latitude winter" for the rest of the year, both included in SBDART [35]. However, these atmosphere models are modified with re-scaled profiles for IWV and ozone to the total IWV (GPS data) and total column ozone (ERA-Interim data-set). The number of streams was set to 4. Additionally, thermal radiation is unset. SW WVRE for solar zenith angle (SZA) > 90° are set to 0 without performing the calculation, since no SW radiation is available under this condition.

The LW simulations, however, used a different configuration. The LW wavelength range was between 4.0 μm and 100 μm (steps of 1% width, that is to say, from 0.04 to 1 μm). The number of streams was set to 16, which is a value used in other works in the LW range [19,36]. The number of atmosphere layers was set to 65, with a resolution between 1 m for the lower layers and 900 m for the higher ones. The atmospheric composition profile from reanalysis is given to SBDART as input, as well as the temperature of the surface from ERA-Interim reanalysis and IWV from GPS (to re-scale the water vapor in the layers). LW calculations have sun radiation unset and thermal radiation activated.

The model was run twice, for each hourly GPS measurement: in one the IWV value from GPS measurement is used, while in the other one, the IWV is set to 0 cm. The irradiances are then used to calculate the WVRE at surface which is defined [13] as the difference between the net irradiances (downwards minus upwards) with and without water vapor in the atmosphere, as shown in Equation (3).

$$\text{WVRE} = \left( I^{\downarrow}_{\text{IWV}} - I^{\uparrow}_{\text{IWV}} \right) - \left( I^{\downarrow}_{\text{dry}} - I^{\uparrow}_{\text{dry}} \right) \tag{3}$$

Total WVRE is defined as the sum of both LW and SW WVRE. The calculated variables are integrated to daily values, in order to compare LW and SW contributions. Missing values are filled with linearly interpolated data, but days with more than 50% missing data are filtered out. The integration is done as an average of the 24 hourly data of each day. It must be noticed that the effects on SW

radiation are limited by insolation, while the effects on LW are active for the whole day, and therefore they must be integrated to daily values before any comparison.

For the study of trends, daily data are deseasonalized, that is to say, we obtained the anomalies. The process is to subtract to every data-point the mean of the data with coincident day of year and site. Then, monthly means of daily deseasonalized data (anomalies) are calculated. Monthly means with 5 days or less are replaced by the linear interpolation at that month. The test used to calculate the trend and decide if it is significant are the Mann–Kendall [37,38] test and Sen's slope [39], since the IWV and WVRE data do not follow a normal distribution. The confidence level considered for significance is 0.05.

## 4. Results

### 4.1. Spatial and Seasonal Variability

In order to study the spatial variability of the WVRE, the sites have been divided into three groups: North Atlantic (NA), Mediterranean Sea (MS) and Interior (I). Figure 1 shows the distribution of stations from each zone in Spain. These groups have a geographical meaning (NA are stations close to the North Atlantic Sea, in Nothern Spain; I stations are in the interior of Iberia, and MS stations are close to the Mediterranean Sea in the East and South of Iberia). This division have been already applied in other water vapor related studies [40,41]. Also, the WVRE exhibits similar features between the sites that belong to the same group.

Table 2 shows some statistics of WVRE by zone and regime. The mean total WVRE is similar in *I* and *NA*, but the *I* zone shows more variability, with longer low-tail (minimum and first quartile are lower than in *NA*) and high-tail (larger maximum and first quartile than in *NA*). MS zone shows generally higher values of total WVRE than the other two zones. Standard deviation (SD) values are around 20 Wm$^{-2}$, while coefficient of variation (CV) values are around 10% for *NA* and 12.5%. The LW regime shows a behavior close to the total regime, with *MS* WVRE values being higher than the other two zones, and *I* and *MS* zones being more disperse than *NA*. Regarding the SW regime, values in the three zones are quite similar. Thus, considering all stations, the mean WVRE is −39.2 Wm$^{-2}$, with a SD of 15.9 Wm$^{-2}$ and a CV of 37.9%. The increased variability (approximately 3 times the LW CV) is due to the seasonality of the solar zenith angle and sunlight hours that heavily affect the SW WVRE.

**Table 2.** Summary statistics of WVRE. All values are in W m$^{-2}$, except CV (in %). The table shows the minimum (min), the first quartile (Q1), the median, the mean, the third quartile (Q3), the maximum (max), the standard deviation (sd) and the coefficient of variation (CV). The zones are: North Atlantic (NA), Mediterranean Sea (MS) and Interior (I).

| Regime | Zone | min | Q1 | Median | Mean | Q3 | max | sd | CV |
|--------|------|-----|-----|--------|------|-----|-----|-----|-----|
| Total | NA | 99.4 | 150.0 | 161.5 | 160.9 | 171.8 | 206.9 | 15.4 | 9.5 |
| Total | MS | 107.6 | 161.5 | 179.9 | 179.5 | 198.0 | 235.8 | 22.4 | 12.5 |
| Total | I | 96.5 | 150.7 | 166.4 | 166.3 | 181.8 | 222.4 | 20.8 | 12.5 |
| Total | All | 96.5 | 156.9 | 173.0 | 174.0 | 192.4 | 235.8 | 22.6 | 13.0 |
| LW | NA | 132.1 | 179.5 | 200.2 | 198.1 | 217.3 | 253.0 | 23.5 | 11.9 |
| LW | MS | 120.5 | 190.6 | 224.0 | 220.2 | 249.8 | 296.7 | 33.9 | 15.4 |
| LW | I | 107.9 | 176.0 | 207.6 | 203.3 | 228.6 | 274.9 | 31.9 | 15.7 |
| LW | All | 107.9 | 185.4 | 214.3 | 213.3 | 242.0 | 296.7 | 33.8 | 15.8 |
| SW | NA | −62.6 | −50.7 | −38.2 | −37.1 | −25.2 | −9.5 | 14.7 | 39.6 |
| SW | MS | −64.9 | −54.2 | −45.2 | −40.6 | −25.0 | −9.8 | 15.1 | 37.2 |
| SW | I | −61.3 | −49.4 | −40.1 | −36.8 | −23.3 | −6.0 | 14.1 | 38.2 |
| SW | All | −64.9 | −52.4 | −42.8 | −39.2 | −24.4 | −6.0 | 14.9 | 37.9 |

Regarding the seasonal behavior, Figure 2 shows a box-plot of the WVRE by months, and Table 3 displays the average values by season: autumn (SON), summer (JJA), spring (MAM) and winter (DJF). It is observed that total WVRE values are quite similar in autumn (SON) as in summer (JJA), while

spring (MAM) values are similar to winter (DJF) ones. Hence, two seasons, for the purpose of total WVRE, could be considered: a cold (spring + winter) season and a warm (summer + autumn) season. For *MS* and *I*, summer total WVRE is slightly over the autumn total WVRE, while in *NA*, the opposite occurs. In Figure 2, it can be observed that SW WVRE is quite similar in the three regions, with the largest effect during summer months ($\sim -50\,\mathrm{Wm}^{-2}$) and the smallest one during winter months ($\sim -20\,\mathrm{Wm}^{-2}$).

**Table 3.** Seasonal mean values of WVRE by regime and zone. Values are in $\mathrm{Wm}^{-2}$. Seasons are winter (December, January, February), spring (March, April, May), summer (June, July, August), and autumn (September, October, November).

| Regime | Zone | Winter | Spring | Summer | Autumn |
|--------|------|--------|--------|--------|--------|
| Total | NA | 151.8 | 147.6 | 164.3 | 173.5 |
| Total | MS | 159.0 | 159.9 | 195.6 | 188.1 |
| Total | I | 144.9 | 151.8 | 179.9 | 171.6 |
| Total | All | 154.4 | 156.4 | 188.8 | 181.3 |
| SW | NA | −16.6 | −38.3 | −53.2 | −31.6 |
| SW | MS | −20.0 | −41.6 | −55.1 | −33.2 |
| SW | I | −17.5 | −37.0 | −50.4 | −29.3 |
| SW | All | −19.1 | −40.0 | −53.5 | −31.7 |
| LW | NA | 168.4 | 186.0 | 217.5 | 205.1 |
| LW | MS | 179.0 | 201.5 | 250.7 | 221.3 |
| LW | I | 162.4 | 188.8 | 230.3 | 200.9 |
| LW | All | 173.5 | 196.4 | 242.2 | 213.0 |

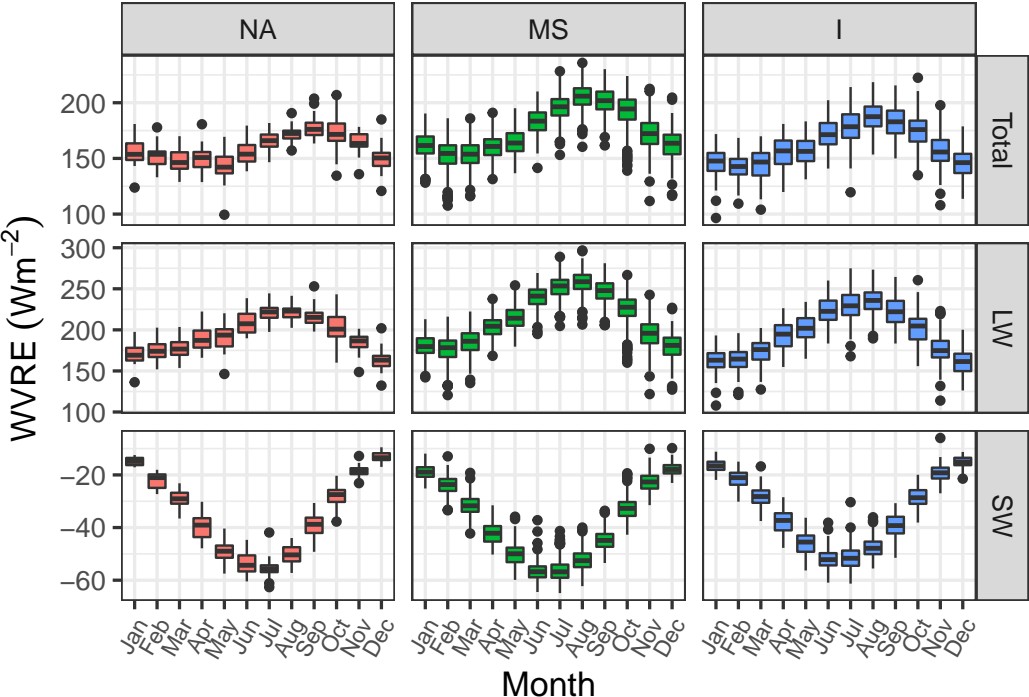

**Figure 2.** Water vapor radiative effects (WVRE) seasonal evolution in the regions considered in this study, in the form of a box-plot.

*4.2. Trends*

Figure 3 shows the evolution of the mean annual IWV and WVRE for the three regimes (LW, SW and Total) on the seven stations. The data have been deseasonalized and then, the monthly averages have been used to build the yearly time series. It is remarkable that, despite the small number of years

(less than a decade) in the time-series, most of the sites and variables show statistically significant results, in all cases with the same sign. IWV has positive trend in all sites, as well as total WVRE and LW WVRE. By contrast, SW WVRE has negative trend, due to the positive trends shown by IWV. Specifically, Table 4 shows IWV and (total, LW and SW) WVRE by stations and zones. IWV trends are about $0.01 - 0.02$ cm year$^{-1}$, statistically significant in all stations but *alme* and *vill*.

**Table 4.** Trends of IWV and WVRE (total, LW and SW). Values are in cm year$^{-1}$ for IWV and Wm$^{-2}$ year$^{-1}$. P-values are shown in parenthesis, and an asterisk (*) is added to statistically significant results.

| Zone | Site | IWV | LW WVRE | SW WVRE | Total WVRE |
|------|------|-----|---------|---------|------------|
| NA | acor | $0.011\ (1.8 \cdot 10^{-03})$* | $0.394\ (1.6 \cdot 10^{-03})$* | $-0.097\ (4.2 \cdot 10^{-04})$ * | $0.348\ (2.9 \cdot 10^{-03})$ * |
| MS | alac | $0.013\ (3.4 \cdot 10^{-03})$ * | $0.199\ (2.6 \cdot 10^{-01})$ | $-0.079\ (6.0 \cdot 10^{-03})$ * | $0.102\ (4.8 \cdot 10^{-01})$ |
| MS | alme | $0.0016\ (8.1 \cdot 10^{-01})$ | $-0.0810\ (7.3 \cdot 10^{-01})$ | $0.0314\ (3.9 \cdot 10^{-01})$ | $-0.0512\ (8.4 \cdot 10^{-01})$ |
| MS | coba | $0.010\ (2.5 \cdot 10^{-02})$ * | $0.377\ (7.7 \cdot 10^{-02})$ | $-0.085\ (2.9 \cdot 10^{-02})$* | $0.346\ (5.6 \cdot 10^{-02})$ |
| MS | vale | $0.0178\ (1.3 \cdot 10^{-03})$ * | $0.1104\ (5.8 \cdot 10^{-01})$ | $0.0095\ (8.1 \cdot 10^{-01})$ | $0.1117\ (4.8 \cdot 10^{-01})$ |
| I | cace | $0.014\ (2.5 \cdot 10^{-03})$ * | $0.770\ (6.4 \cdot 10^{-04})$ * | $-0.103\ (6.7 \cdot 10^{-03})$ * | $0.707\ (4.0 \cdot 10^{-04})$ * |
| I | vill | $0.0059\ (1.2 \cdot 10^{-01})$ | $0.5005\ (2.4 \cdot 10^{-02})$ * | $-0.0399\ (2.8 \cdot 10^{-01})$ | $0.4230\ (2.6 \cdot 10^{-02})$ * |

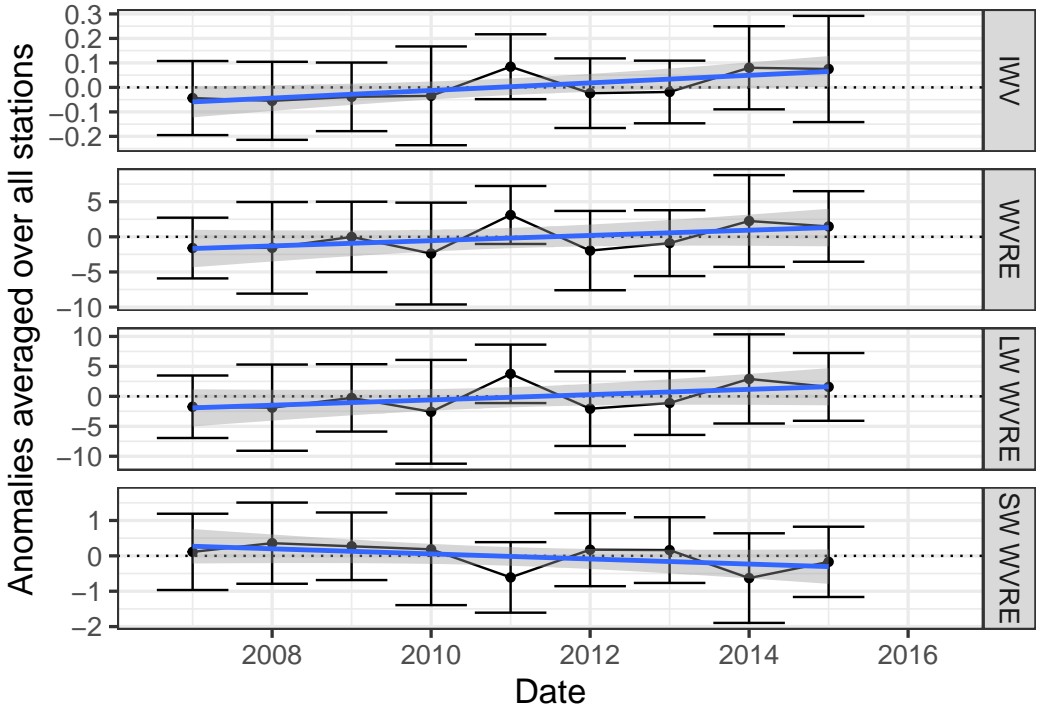

**Figure 3.** Time-series of station-averaged integrated water vapor (IWV) and WVRE (SW, LW and total regimes) anomalies. Values are in cm for IWV and in Wm$^{-2}$ for WVRE. Blue solid lines point out the linear trends (see text).

## 5. Discussion

### 5.1. Spatial and Seasonal Variability

Mateos et al. [23] reported an averaged value of $-57.1$ Wm$^{-2}$ (SD of 16.2 Wm$^{-2}$) for the combined radiative effect caused by clouds and aerosols over the Iberian Peninsula for the period 1985–2010. Additionally, Mateos et al. [22] derived the aerosol radiative effect under cloud-free conditions at six stations located in the Iberian Peninsula, reporting annual averages in the range of $-8.8$ to $-5.7$ Wm$^{-2}$. Hence, the comparison of these radiative effects with the results reported in Figure 2 and Table 3 for

the WVRE points out the great influence of the water vapor on the SW irradiance in the study region. Moreover, the large differences between seasons shown in Figure 2 can be related to the marked seasonal cycle of water vapor column over the Iberian Peninsula [42], in which IWV is larger in summer and smaller in winter.

The differences among regions are more noticeable in the LW range, as well as in the total WVRE (because of the LW contribution). Region *I* shows values below the *MS* region, but a similar seasonal variability. However, the *NA* region shows a less marked seasonal variability, being the difference between winter and summer more subtle than in the other two cases. Therefore, NA winter LW WVRE is similar to *MS* winter LW WVRE, while *NA* summer LW WVRE is similar to *I* summer LW WVRE.

*5.2. Trends*

The trend values reported in Table 4 for IWV are about three times higher than those obtained by Ning and Elgered [43] for Sweden and Finland, using GPS data during the period 1997–2016 ($\sim 0.004$ cm year$^{-1}$). Chen and Liu [44] showed values around 0.002 cm year$^{-1}$ in temperate latitudes for the period 2000–2014, one order of magnitude below our results. Vicente-Serrano et al. [45] also detected positive trends in specific humidity at surface in Spain in the period 1961–2011.

The positive trends in IWV cause the total WVRE trend to be significantly positive in stations from regions *I* and *NA*, while non-significant in *MS* stations. The LW WVRE trends are positive as well (except for *alme*, with a non-significant negative trend), while the SW WVRE trends are negative (except for *alme* and *vale* both with positive but non-significant trends). The balance between both LW and SW WVRE gives a positive trend (except for *alme*, with a non-significant negative trend). The SW WVRE significant trends are around $-0.09$ Wm$^{-2}$ year$^{-1}$, while the LW WVRE significant trends are around 0.50 Wm$^{-2}$ year$^{-1}$. Then the overall significant trends on WVRE are around 0.42 Wm$^{-2}$ year$^{-1}$. This positive trend can partially explain the rise in surface air temperatures observed in the Iberian Peninsula during the last two decades (e.g., [46]), which in turn increase the evaporation in a positive climate feedback [3].

The mean SW WVRE trend values are weaker ($-0.09$ Wm$^{-2}$ year$^{-1}$) than Kvalevåg and Myhre [47], where global SW trends due to water vapor are estimated as $-0.29$ Wm$^{-2}$ per year. This result indicates that water vapor could have a role in modulating the widespread increase of SW surface radiation, also known as *brightening*, reported in the literature since the 1980s [48,49]. Mateos et al. [24] determined that this SW radiation trend is $+0.7$ Wm$^{-2}$ year$^{-1}$ on average for the period 2003–2012 in the Iberian Peninsula, using both ground-based and satellite SW data. These authors also showed that three fourths of the trend is explained by clouds, while the other one fourth is related to aerosol change, in line with the observed reductions in total cloud cover and aerosol load over the study region. Additionally, Mateos et al. [22] reported a statistically significant trend of $+0.36$ Wm$^{-2}$ year$^{-1}$ for the aerosol radiative effect under cloud-free conditions in the Iberian Peninsula (period 2004–2012). Hence, it must be pointed out that the negative trend for SW WVRE is about a quarter of this positive trend for the aerosol radiative effect. Therefore, the trends of water vapor could be partially masking the full magnitude of the role of aerosol load in the modulation of SW radiation at surface over the study region.

## 6. Conclusions

This work has provided some insight about the radiative effects of the water vapor in Sothwestern Europe in both the long wave and short wave bands under cloud and aerosol load free conditions. The results show that the three regions considered in the Iberian Peninsula have total WVRE around 173 Wm$^{-2}$, with total maximum of 235.8 Wm$^{-2}$ and minimum of 96.5 Wm$^{-2}$. The LW WVRE is therefore larger in absolute value than the SW WVRE. Specifically, LW WVRE values range from 107.9 Wm$^2$ to 296.7 Wm$^{-2}$, while SW WVRE exhibits negative values, from $-64.9$ Wm$^{-2}$ to $-6.0$ Wm$^{-2}$.

The distribution of the WVRE has a marked seasonal cycle in all zones considered. In general, LW WVRE is higher in summer and autumn, and lower in winter and spring. However, the SW WVRE has stronger values (more negative) in spring and summer, and weaker in autumn and winter. Overall, the total WVRE follows the LW WVRE pattern, with stronger values in summer and autumn, and weaker WVRE in winter and spring.

The trends have been calculated for IWV and WVRE in the total, LW and SW regimes. IWV trends are positive in all cases, with a mean value of $0.013$ mm year$^{-1}$, and this causes the LW and total WVRE trends to be positive (mean values of 0.56 and 0.49 Wm$^{-2}$year$^{-1}$, respectively) and SW WVRE trends to be negative ($-0.09$ Wm$^{-2}$year$^{-1}$).

It must be highlighted that the positive radiative effect in the whole spectral range, associated with the increase of the water vapor over the Iberian Peninsula, may partially explain the notable increase of the surface air temperature reported in the literature in this region. Additionally, the negative radiative effect in SW, due to the notable increase of IWV values over the Iberian Peninsula during the last decade, may play a key role in mitigating the SW radiation increases associated with a reduction of the cloud cover and aerosol load over this region. Therefore, this increase of the water vapor could partially offset the strong brightening effect (increase of SW radiation at surface) recorded in the Iberian Peninsula since 2000s.

**Author Contributions:** J.V.-M. performed the data analysis and wrote the main draft of the paper. M.A. provided the main ideas and contributed to the data analysis and paper writing. A.S.-L. contributed to the discussion of results and paper writing. V.E.C. provided GPS, temperature, pressure and sunshine duration data and contributed to paper writing. All authors have read and agreed to the published version of the manuscript.

**Funding:** This work was partly supported by the Ministerio de Economia y Competitividad of the Spanish Government (CGL2017-87917-P) and by Junta de Extremadura and FEDER funds (IB18092). V.E.C. is grateful to the Spanish Ministry of Science, Innovation and Universities for the support through the ePOLAAR project (RTI2018-097864-B-I00). A.S.-L. was supported by a postdoctoral fellowship RYC-2016–20784 funded by the Spanish Ministry of Science, Innovation and Universities. J.V.-M. was supported by a predoctoral fellowship (PD18029) from Junta de Extremadura and European Social Fund.

**Acknowledgments:** Authors thank AEMet and ECMWF for providing the data necessary for this work, and to Paul Ricchiazzi for the SBDART radiative transfer code. Authors also acknowledge the R [50] packages tidyverse 1.2.1 [51], lubridate 1.7.1 [52], ggpubr 0.2.4 [53], zoo 1.8-7 [54], trend 1.1.2 [55].

**Conflicts of Interest:** Authors declare no conflict of interest.

## Abbreviations

| | |
|---|---|
| IWV | Integrated Water Vapor |
| WVRE | Water Vapor Radiative Effects |
| SW | short-wave |
| LW | long-wave |
| GPS | Global Positioning System |
| NA | North Atlantic zone |
| I | Interior zone |
| MS | Mediterranean Sea zone |
| ZTD | Zenith Tropospheric Delay |
| ZWD | Zenith Wet Delay |
| ZHD | Zenith Hydrostatic Delay |
| EUREF | European Reference Frame |
| AEMet | Spanish Meteorological State Agency |
| sd | standard deviation |
| CV | Coefficient of Variation |

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
