# Peer review of "Evaluation of Water Vapor Radiative Effects Using GPS Data Series over Southwestern Europe"

_remotesensing, doi:10.3390/rs12081307_

Round 1

Reviewer 1 Report

These comments concern manuscript 767991 whose title is:  “Evaluation of water vapor radiative effects using GPS data series over Southwestern Europe”.  The research described uses atmospheric water vapor measurements made through the use of global positioning satellite signals to determine LW and SW radiances through simulations.  The simulations use either the water vapor from the GPS measurements or is set to zero – these two are then compared to determine the purported impact of water vapor on the radiances.  Changes in the impact of water vapor over time are then used to “explain” surface temperature increases during the study period.

The paper is reasonably well written and organized; however, there are several issues related to use of “English” which lead to “awkward” phrases and, perhaps, improper use of English.  I will attempt to list some; but, not all of these issues:

line 6 – “… goes from…” would probably be better replace by “ranges from”

line 7 – “to classify” would probably be better as “the classification of”

line 22 – “…what allows heating of the low layer of air…” might be better as “…which allows heating of the lower atmosphere…” or “…which allows heating of the lowest layer of the atmosphere…”

line 43 – “of climate” should be “of the climate”

of the paragraph are those references, the it needs to be stated that the previous studies are those references…

Figure 1 – in my printed copy, the names of the I and MS stations are the same color as are the “dots” showing location…

line 77 -- “close” should probably be “nearby”…

line 82 – “in” should probably be “for”…

line 85 – “Some data”  should be replaced by “Some additional data”

line 105 – in my printed copy, I find several instances of “Ì„m ”  -- a superscripted minus after the m – not sure what is going on except the “units” are incorrect…

line 126 – “between” should be “of”

Figure 2 – need a description of the symbols on the plots and the especially the “dots”…

line 140 – the sentence which begins on this line is awkward and meaning unclear…

line 149 – meaning of “…being more spread…” is unclear…

line 153 – most of the paragraph contains awkward and unclear phrases…

line 169 – “statistically” should probably be replaced by “which are statistically”

line 199 – I am not sure that what was reported by reference 43 was a “notable increase in surface air temperature…”.  What they did report was:  “Significant linear trends observed for T m, T x, and T n versus year were 56, 58, and 47 % of the weather stations, respectively, with temperature ranges between 0.2 and 0.4 °C per decade.”  I think that use of the term “notable” is too strong when ~50 % of the stations did not show significant trends…

line 206 – “obtained” should probably be “determined”…

Author Response

Dear Reviewer 1,

Thank you very much for your detailed list of corrections, it is definetively improving our manuscript. We have accepted all your suggestions.

As for this phrase "of the paragraph are those references, the it needs to be stated that the previous studies are those references…" we do not really understand what you mean.

The greek letter "mu" was not properly formated for "micrometers" units. This has been corrected.

Figure 2 are boxplots and this information has been added to the caption. In boxplots, outliers are typically indicated as dots.

From line 199, we eliminated the word "notable"

Reviewer 2 Report

See comments attached.

Author Response

Dear Reviewer 2,

Thank you for your comments. They have improved our manuscript very much. We agree and have changed accordingly all your suggestions.

We have added an "abbreviations" with the abreviations and acronyms used in the manuscript.

The reason for the time range (explained in the new version of the manuscript) is that this is when we have data available for the GPS stations.

The reason for 2011 being different from other years is simply that it was significantly wetter, with more IWV, and therefore, stronger values of WVRE in LW, SW and total (LW+SW).

The process to obtain the anomaliles is explained at the end of methodology section. However, we did not do any reference to the anomalies. So, to clarify this, we have added the word "anomalies" in the text, in order to clarify this aspect.